# Persuade with Reason: Enhancing Debate Persuasiveness through Accurate Persuasion Feedback Derived from Weak Supervised Labels

## ABSTRACT

Existing methods for debate generation often struggle to provide convincing proof, lacking critical persuasiveness. More challengingly, directly fine-tuning or using RLHF on large language models (LLMs) can decrease the persuasiveness of the generated text, making it difficult to leverage advancements from state-of-the-art LLMs. We identify two key biases underlying this issue: reward hacking and reward sparsity. Reward hacking blurs the model's training objectives, causing the model to focus more on linguistic style and rhetoric while neglecting the essential logical reasoning and value shaping. Reward sparsity reduces the generalization and robustness of the reward model. To address these two problems, we propose a novel persuasiveness enhancement training method: $P^3$. Firstly, we introduce **P**ersuasive reward estimation and modeling by separating persuasiveness scores from surface cues, addressing the reward hacking problem. Secondly, we solve the reward sparsity issue by employing **P**ersuasive sample mining to extract persuasive annotation information from weakly supervised labels. Lastly, we design a new DPO algorithm tailored for **P**ersuasiveness generation optimization, which modifying the objective function to mitigate the divergence problem on debate generation task. Extensive experimental results demonstrate that $P^3$ effectively alleviates the aforementioned issues, significantly enhancing the model's performance in debate and persuasion tasks, surpassing state-of-the-art closed-source commercial models, such as Gemini and Claude, in both automatic and human evaluations.

## 1 INTRODUCTION

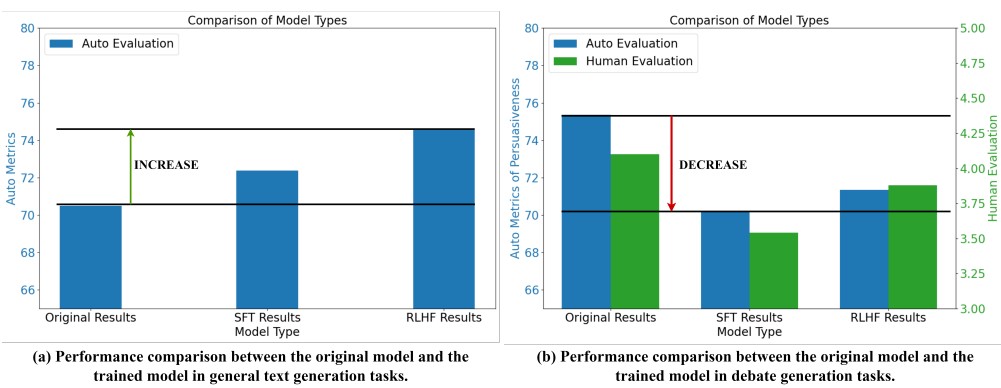

(a) Performance comparison between the original model and the trained model in general text generation tasks.

(b) Performance comparison between the original model and the trained model in debate generation tasks.

Figure 1: Comparison of model performance before and after training in debate generation tasks and general text generation tasks.Unlike general text generation tasks, the performance of debate generation tasks decreases after applying SFT or RLHF.

Persuading someone to change their viewpoint is a common goal across various applications, from political campaigns and marketing to competitive debates. The task of debate and persuasion generation aims to create text that convinces a specific audience. This task is prominent and has been the focus of substantial research efforts (Cialdini & Cialdini, 2007; Petty & Cacioppo, 2012). However, existing methods, while capable of producing meaningful viewpoints, often struggle to

provide convincing proof (Xiao et al., 2024). Generated articles may present contradictory evidence or fail to combine assertions into a logical flow, resulting in text that lacks persuasiveness. Additionally, whether fine-tuning large language models (LLMs) directly or using reinforcement learning with human feedback (RLHF), the persuasiveness of the generated text tends to decline, making it challenging to leverage advancements in state-of-the-art LLM technologies for this task (refer to Figure 1).

Through further analysis of error samples, we identified that the decline in model performance post-training is due to two critical issues: reward hacking and reward sparsity.

Reward hacking arises from a significant deviation between training objectives and actual persuasiveness. As shown in Figure 2(a), text similarity metrics (such as BLEU and ROUGE) and the trained reward model diverge considerably from the persuasiveness scores given by human annotators. Linguistic research indicates that persuasive arguments rely on well-defined claims, sound reasoning, and credible evidence, reinforced through rhetoric (Hubbart, 2025). However, our analysis reveals that existing training objectives emphasize superficial cues such as language structure and style, while neglecting deep semantic elements like logical validity, leading to inaccurate persuasiveness score feedback. For example, in Figure 2(b), generated text using phrases like "For instance" to provide an example is deemed persuasive, even if the example contradicts the core thesis. Consequently, both SFT methods (with the training objective being similarity to ground truth) and RLHF methods (with the training objective being maximization of reward scores) fail to optimize for persuasiveness correctly. This results in models disproportionately learning debate language style and rhetorical techniques, while overlooking crucial logical reasoning and value shaping.

Reward sparsity arises from the scarcity of persuasive data. Most natural language processing tasks and standard evaluation sets lack the process of claim-challenging-persuading the audience. Typically, tasks only contain a question and a standard answer, failing to capture debate and persuasion dynamics. Technical reports on large language models such as LLaMA (Touvron et al. (2023a;b)) indicating that over 90% of supervised fine-tuning (SFT) data consists of a single stance or answer. Even in the few available online debate and persuasion datasets, persuasiveness labels are sparse. For example, in the ChangeMyView (CMV) dataset (Tan et al., 2016), less than 1% of the training samples have accurate persuasiveness labels ($\Delta$). The lack of sufficient data leads to inadequate training of the reward model, making it challenging to effectively evaluate new responses generated by the model during RLHF processes.

To address the aforementioned issues, we propose a novel persuasiveness-enhanced training method: $P^3$, which consists of three comprehensive stages: (1) **P**ersuasiveness Reward Estimation and Modeling: This stage primarily addresses the reward hacking problem. We model the debate persuasion process as a Markov decision process and use action-value functions to estimate accurate persuasiveness and superficial cues scores. This approach guides the model to focus on core elements of persuasiveness. (2) **P**ersuasiveness Sample Mining: This stage tackles the reward sparsity issue. We use the difference between upvotes and downvotes as weak supervision labels to address the lack of precise persuasiveness labels ($\Delta$) in datasets like CMV. To avoid introducing noise, we use the persuasiveness scores extracted in stage (1) to identify high-quality debate response. (3) **P**ersuasiveness Strategy Optimization: In this stage, we design a new offline DPO algorithm named PAPO. The algorithm optimizes the objective function to ensure the model focuses on the accurate persuasiveness scores, and avoids overfitting to noise in the weak supervision labels.

Experimental results on the CMV dataset demonstrate that $P^3$ significantly enhances the persuasiveness performance of the base model on both automated and human evaluation metrics. Moreover, using a smaller base model (13B parameters), $P^3$ surpasses state-of-the-art closed-source commercial models such as Gemini and Claude.

In summary, our method offers the following contributions:

- We model the debate persuasion process as a Markov decision process and use action-value functions to accurately separate persuasive elements from superficial cues, addressing the reward hacking problem in debate generation.

- We use weak supervision labels for persuasiveness sample mining, tackling the issue of insufficient precise persuasiveness labels and alleviating the reward sparsity problem.

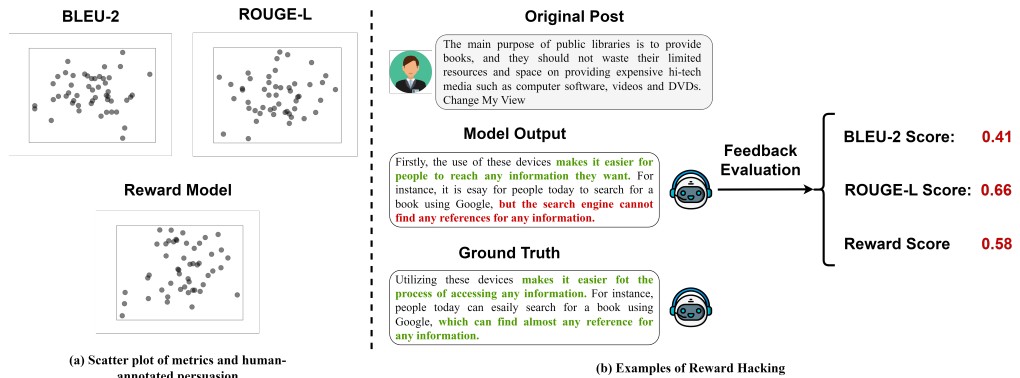

(a) Scatter plot of metrics and human-annotated persuasion

(b) Examples of Reward Hacking

Figure 2: The provided examples illustrate the impact of reward hacking. The scatterplot and examples indicates a significant deviation between training objectives and actual persuasiveness.

- We design a novel PAPO algorithm, modifying the DPO objective function specifically for the debate generation scenario. Experimental results show that our method, using a smaller base model, outperforms state-of-the-art closed-source commercial models.

## 2 METHOD

### 2.1 OVERVIEW

To address the issues of reward hacking and reward sparsity, we designed three main components. Firstly, Persuasiveness Reward Estimation and Modeling uses maximum likelihood and the EM algorithm to separate accurate persuasiveness scores from surface cue scores, addressing reward hacking issues. Subsequently, Persuasiveness Sample Mining extracts samples dominated by persuasive elements from crowdsourced weakly supervised data, resolving the sparsity problem of human annotated $\Delta$ persuasiveness labels. Finally, the improved PAPO algorithm ensures that the model focuses on precise persuasiveness scores and avoids overfitting to the noise present in weakly supervised labels, overcoming the drawbacks of the original DPO algorithm.

### 2.2 TASK DEFINITION

The debate generation task can be formally defined as follows: Given an original post $x_p$ that ends with 'Please Change My View,' indicating that the OP is inviting other users to debate and reach a persuasive conclusion, the task involves training a model to generate persuasive debate text. Within the discussion thread of this post, some users have already engaged in historical discussions with the OP, defined as $x_h$. The goal of debate generation is to train a model $\mathcal{M}_\theta(x_p, x_h)$ to generate persuasive debate text $Y$ based on the original post and historical discussions, i.e., $\mathcal{M}_\theta(x_p, x_h) = Y$.

### 2.3 PERSUASIVENESS REWARD ESTIMATION AND MODELING

#### 2.3.1 PERSUASIVENESS MODELING

To address the scarcity of precise persuasiveness labels ($\Delta$) in the CMV dataset (Tan et al., 2016), we use "scores" ($s$) as a weak supervision signal to reflect each post's persuasiveness level. This metric is defined as the difference between the number of upvotes and downvotes for each post. It serves as a crowdsourced annotation of post quality by Reddit users and reflects the persuasiveness of the post in relation to the OP's viewpoint. However, these scores contain substantial information unrelated to persuasiveness. Users might be influenced by the literal quality of the text, such as vocabulary richness and sentence structure, causing a high score to be attributed either to high logical persuasiveness or superior literal quality. This leads to significant reward hacking issues when directly using the score to train a reward model, making it difficult to provide meaningful guidance for model

optimization. To clarify the reasons behind each post's score, this paper models each post's score using a Bernoulli distribution and separately calculates persuasiveness scores and literal scores.

Each post is assigned two scores: a persuasiveness score and a literal score. The literal score $s_s$ is based solely on the text's intrinsic features, such as vocabulary and grammar, independent of the debate logic between the parties. Thus, the literal score $s_s$ is defined as a function of the generated debate text $\hat{y}$:

$$s_s = f_s(\hat{y}) \tag{1}$$

In contrast, the persuasiveness score $s_d$ is determined by the logical interaction between the texts of both parties and the current environment, making it a ternary function:

$$s_d = f_d(\hat{y}, x_p, x_h) \tag{2}$$

where $x_p$ is the post of the OP, $\hat{y}$ is the generated debate post, and $x_h$ is the historical speech of other users.

Given the varying focus points of the scoring human audience, the observed score $s$ might depend on either the persuasiveness score or the literal score, following a Bernoulli distribution:

$$p(s \mid \hat{y}, x_p, x_h) = \begin{cases} \alpha, & s = f_d(\hat{y}, x_p, x_h) \\ 1 - \alpha, & s = f_s(x) - f_s(x_p) \end{cases} \tag{3}$$

where $\alpha$ is the prior probability that the human audience emphasizes the persuasiveness score.

Inspired by Du et al. (2024) and Du et al. (2023), in order to derive the calculation methods for each score from a dataset annotated by humans, we use two MLPs (Multi-Layer Perceptrons) to fit the persuasiveness score and the literal score, respectively:

$$f_d(\hat{y}, x_p, x_h) = f_d(\hat{y}, x_p, x_h; \theta_d) \tag{4}$$

$$f_s(x) = f_s(x; \theta_s) \tag{5}$$

where $\theta_d$ and $\theta_s$ are the parameters of the neural networks.

After defining the hybrid debate score model, relative action-value function estimation is employed to train the model, resulting in the accurate calculation of both scores for each post.

### 2.3.2 PERSUASIVENESS REWARD ESTIMATION

To ensure the practical significance of the score, this paper models posts, historical posts, posting actions, and the upvote-minus-downvote count of each post as the environment, action, and reward in a Markov Decision Process (MDP), respectively. We then use the action-value function to describe the persuasiveness score $s_d$.

However, it is impossible to use temporal-difference (TD) (Sutton, 1988) or Monte Carlo methods (Metropolis & Ulam, 1949) to directly fit the action-value function within the hybrid debate score model. First, since the score $s$ is sampled from both the literal score and the persuasiveness score, any TD or mean squared error (MSE) calculated using $s$ would be non-differentiable, making optimization of model parameters infeasible. Second, due to the varying popularity of topics, posts on popular topics receive significantly more upvotes than those on less popular ones, meaning the reward value does not directly indicate persuasiveness. Only the relative reward size within the same topic can accurately describe persuasiveness strength. Therefore, we indirectly fit the persuasiveness score by approximating the win rate defined by the Bradley-Terry model Bradley & Terry (1952).

Specifically, the win rate of post $\hat{y}^{(1)}$ over $\hat{y}^{(2)}$ in the same context depends on the difference in their scores, represented as $\sigma(s^{(1)} - s^{(2)})$. Assuming that whether each post's score is based on persuasiveness is independent, the event $y$ that post $\hat{y}^{(1)}$ wins follows a mixed Bernoulli distribution:

$$\begin{aligned} p(y) &= \sum_{s^{(1)} \in \{s_d^{(1)}, s_s^{(1)}\}} \sum_{s^{(2)} \in \{s_d^{(2)}, s_s^{(2)}\}} p(s^{(1)}, s^{(2)}) \sigma(s^{(1)} - s^{(2)}) \\ &= \alpha^2 \sigma(s_d^{(1)} - s_d^{(2)}) + \alpha(1 - \alpha)(\sigma(s_d^{(1)} - s_s^{(2)})) + \alpha(1 - \alpha)(\sigma(s_s^{(1)} - s_d^{(2)})) \\ &\quad + (1 - \alpha)^2 \sigma(s_s^{(1)} - s_s^{(2)}) \end{aligned} \tag{6}$$

In the aforementioned model, the observed win rate is a probabilistic parameter model containing hidden variables $s_s$ (persuasiveness score) and $s_d$ (literal score). This can be solved using the EM algorithm and maximum likelihood estimation (MLE) (Dempster et al., 1977). In the E-step of each iteration, the distribution $q$ represents the posterior distribution of the scores for both posts:

$$q(s^{(1)}, s^{(2)}) = p(s^{(1)}, s^{(2)}|y) = \frac{p(s^{(1)}, s^{(2)})\sigma(s^{(1)} - s^{(2)})}{p(y)} \qquad (7)$$

In the M-step, the goal is to maximize the following objective function:

$$L(s^{(1)} > s^{(2)}) = \sum_{s^{(1)} \in \{s_d^{(1)}, s_s^{(1)}\}} \sum_{s^{(2)} \in \{s_d^{(2)}, s_s^{(2)}\}} q(s^{(1)}, s^{(2)}) \log\left(p(s^{(1)}, s^{(2)})\sigma(s^{(1)} - s^{(2)})\right) \qquad (8)$$

where $s^{(1)}$ is the winner among each pair of posts.

To determine the winner within each pair in the dataset, the Bradley-Terry model is used again. The win rate is derived from their discounted cumulative rewards, and the expected loss when each post wins is computed based on the win rate, forming the final objective function:

$$E_D\left[\sigma(g^{(1)} - g^{(2)})L(s^{(1)} > s^{(2)}) + \sigma(g^{(2)} - g^{(1)})L(s^{(2)} > s^{(1)})\right] \qquad (9)$$

where $D$ is the dataset, and $g^{(1)}$ and $g^{(2)}$ are the discounted cumulative rewards of the two posts.

After training the hybrid debate score model using relative action-value function estimation, we can obtain both scores for each post, allowing us to filter out samples dominated by literal scores and retain those led by persuasiveness scores.

## 2.4 Persuasiveness Sample Mining

The LLM has already undergone extensive training on literal scores during the SFT stage, making further training with samples dominated by literal scores from the persuasiveness dataset unnecessary. Therefore, for subsequent persuasiveness-enhancement training, we will only select samples dominated by persuasiveness scores $s_d$.

Given that the hybrid debate score model may include some errors, directly using the persuasiveness scores of all samples for training could introduce noise. To avoid this, we will select the longest sequence $L$ in each scenario where the order of original scores ($s$) matches the order of persuasiveness scores ($s_d$), and use these as the persuasive samples for later training:

$$L = \text{argmax}_D |D| \quad \text{s.t.} \quad \forall i, j \in D, \quad \text{sign}(s^{(i)} - s^{(j)}) = \text{sign}(s_d^{(i)} - s_d^{(j)}) \qquad (10)$$

The selection is solved using a dynamic programming algorithm, with the specific process shown in Algorithm 1 (refer to Appendix A). The time complexity is $O(n \log n)$.

## 2.5 Persuasiveness Strategy Optimization

Due to the absence of real-time human audience for scoring debates, we utilized an offline DPO algorithm to enhance the persuasiveness of the LLM. However, traditional DPO (Rafailov et al., 2023) may cause the LLM strategy to diverge when used on small sample datasets. For instance, when the dataset contains only one preference pair. To simplify the notation, we define:

$$r(y) = \frac{\pi(y|x_p, x_h)}{\pi_0(y|x_p, x_h)} \qquad (11)$$

the DPO loss

$$E_D\left[\log \sigma\left(\beta \log\left(\frac{r(\hat{y}^{(1)})}{r(\hat{y}^{(2)})}\right)\right)\right] \qquad (12)$$

results in a constant positive gradient for the probability of the winner's post $\pi(\hat{y}^{(1)}|x_p, x_h)$:

$$\beta \left(1 - \sigma \left(\beta \log \left(\frac{r(\hat{y}^{(1)})}{r(\hat{y}^{(2)})}\right)\right)\right) \frac{1}{\pi(\hat{y}^{(1)}|x_p, x_h)} \tag{13}$$

Consequently, $\pi(\hat{y}^{(1)}|x_p, x_h)$ will eventually approach 1, leading to overfitting. To address this issue, we introduced Persuasion Augment Policy Optimization (PAPO), adding a smoothing term coefficient based on the persuasiveness scores into DPO loss. The improved training objective can be defined as:

$$E_D \left[\sigma \left(s_d^{(1)} - s_d^{(2)}\right) \log \sigma \left(\beta \log \left(\frac{r(\hat{y}^{(1)})}{r(\hat{y}^{(2)})}\right)\right) + \sigma \left(s_d^{(2)} - s_d^{(1)}\right) \log \sigma \left(\beta \log \left(\frac{r(\hat{y}^{(1)})}{r(\hat{y}^{(2)})}\right)\right)\right] \tag{14}$$

This approach effectively prevents the DPO divergence issue on small sample datasets compared to directly using persuasiveness scores to generate preference pairs, followed by DPO. The gradient of our method is:

$$\sigma \left(s_d^{(1)} - s_d^{(2)}\right) \beta \left(1 - \sigma \left(\beta \log \left(\frac{r(\hat{y}^{(1)})}{r(\hat{y}^{(2)})}\right)\right)\right) \frac{1}{\pi(\hat{y}^{(1)}|x_p, x_h)}$$
$$-\sigma \left(s_d^{(2)} - s_d^{(1)}\right) \beta \left(1 - \sigma \left(\beta \log \left(\frac{r(\hat{y}^{(1)})}{r(\hat{y}^{(2)})}\right)\right)\right) \frac{1}{\pi(\hat{y}^{(1)}|x_p, x_h)} \tag{15}$$

The sign is not constant, ensuring that $\pi(\hat{y}^{(1)}|x_p, x_h)$ does not diverge. Additionally, by finding the stationary point of the objective function, our method ensures that the LLM strategy converges to the optimal PPO (Proximal Policy Optimization) solution,

$$\pi(\hat{y}|x_p, x_h) \propto \pi_0(\hat{y}|x_p, x_h) \exp \left(\frac{1}{\beta} s_d\right) \tag{16}$$

thus providing good interpretability.

## 3 EXPERIMENTAL SETTING

### 3.1 DATASET

The ChangeMyView (CMV) dataset is derived from the /r/ChangeMyView subreddit, which boasts over 211,000 users. In this forum, an original poster (OP) shares their viewpoint on a specific topic and invites users to respond in an attempt to change their perspective, known as a "Change My View" request. If a user successfully persuades the OP, they receive a mark ($\Delta$) indicating the OP has been convinced. The forum is dedicated to civil discourse, with moderators and administrators enforcing rules to ensure thorough expression of perspectives during debates. The CMV dataset contains over 1,000,000 discussion nodes and 60,000 unique users, with detailed statistics and examples available in Appendix B. Due to its large volume of data and high-quality persuasive debates, the ChangeMyView dataset has become a benchmark for debate and persuasion generation tasks.

### 3.2 EVALUATION METRICS

**Automated Evaluation Metrics.** As indicated in Section 1 and Figure 1, traditional word overlap-based automated evaluation metrics (e.g., BLEU, Rouge) significantly diverge from the true persuasiveness scores. Although closed-source commercial models such as OpenAI opt for human evaluation for the CMV task (Jaech et al., 2024), financial constraints prevent us from performing manual evaluation over the entire test set. Multiple studies have demonstrated that using GPT-4 for open-domain text generation evaluation greatly enhances the consistency between automated metrics

Table 1: The results of comparison of baselines on automatic metrics and human evaluation metrics. For automatic evaluation metrics, we perform non-replacement sampling 3 times on the test set, each time sampling 10%, and report the average results. For human evaluation metrics, we sample 100 instances for assessment. † means statistically significant difference (2-tailed t-test, $p<0.05$). **Bold** numbers denote the best performance among all methods.

| Method | Base Model Type | #Params | o1-score | Human Evaluation |
|---|---|---|---|---|
| Qwen2-13B | Community | 13B | 73.57 | 4.10 |
| Qwen2-72B | Community | 72B | 75.73 | 4.32 |
| Gemini1.5 Flash | Commercial | 175B | 77.01 | 4.60 |
| Claude3 Haiku | Commercial | 175B | 77.85 | 4.58 |
| **Ours** | Community | 13B | **78.29**† | **4.62**† |

and human evaluation (Hu et al., 2023; Liu et al., 2023; Fu et al., 2023). Therefore, we employ GPT4-o1 to simulate human-like persuasiveness scoring, referred to as o1-score. Further analysis (refer to Figure 3) shows that this automated metric has a high Pearson correlation coefficient (0.67) with human evaluation scores, accurately reflecting the level of text persuasiveness. The specific prompts and settings used for the evaluation are detailed in Figure 3.

**Human Evaluation Metrics.** Based on the evaluation approach outlined in OpenAI's GPT4-o1 technical report for the CMV task (Jaech et al., 2024), we conducted sampled human evaluation of the dataset. Specifically, we recruited three proficient English speakers with debate backgrounds to manually evaluate the generated outcomes. We established the following two evaluation tasks:

(i) *Scoring the persuasiveness of the generated text.* Similar to o1-score, annotators scored the extent to which the generated text persuaded them on a scale of 0-5.

(ii) *Comparing the results to baselines.* Annotators compared the outputs of our proposed method against all baseline results, providing a *Win*, *Loss*, or *Tie* judgment for each pair of test samples.

### 3.3 Implementation Details

We train the base model with the help of huggingface, DeepSpeed and trlx. The base model of our approach is Qwen2-13B. We train the model in 5 epochs. The batch size per device is set to 8. All experiments are conducted with NVIDIA Tesla A100 GPU.

## 4 Results and Analysis

### 4.1 Main Results

**Performance on Automatic Evaluations.** As shown in Table 1, the automatic evaluation results indicate that our method mitigates reward hacking and reward sparsity during training, significantly enhancing the persuasiveness of model-generated texts. Specifically, compared to mainstream open-source models (Qwen2-13B, Qwen2-72B), our method increases the o1-score by an average of 3.6 points. Additionally, when compared to closed-source commercial models, our approach achieves better persuasiveness with fewer parameters and training epochs, increasing the o1-score by an average of 0.9 points.

**Performance on Human Evaluations.** Table 1 and Figure 4 present the human evaluation metrics. As shown in Table 2, human evaluation results demonstrate that our method produces texts that are more persuasive to human reviewers compared to various strong baseline methods. Specifically, our method improves human evaluation metrics by an average of 0.40 points over open-source models. When compared to closed-source commercial models, the generated texts remain competitive, showing an improvement of 0.03 points on average. Figure 2 illustrates the direct comparison results between our method and various strong baselines. The results indicate that our method significantly outperforms mainstream open-source models, with 64% of the generated results surpassing baseline models on average. Compared to closed-source commercial models, our approach also shows a notable advantage, with 34% of the results surpassing baseline models and 80% being comparable

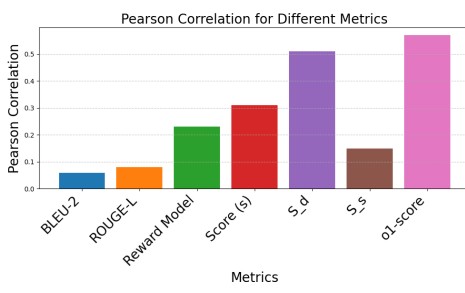

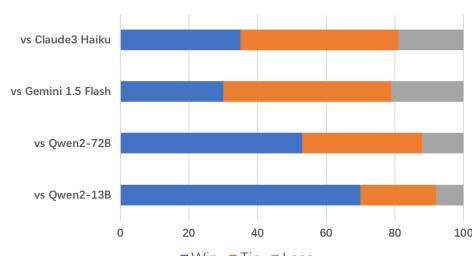

Figure 3: Figure illustrating the Pearson correlation coefficients between various evaluation metrics and human-annotated persuasiveness. Here, $S_d$ represents the persuasiveness score, and $S_s$ represents the superficial cue score.

Figure 4: $P^3$ compared to other baselines. Human raters compared different model generations and and scored them accordingly.

Table 2: The results of ablation experiments. **Bold** numbers denote the best performance.

| Method | o1-score | Human Evaluation |
|---|---|---|
| **Ours** | **78.29** | **4.62** |
| *-w/o Persuasiveness Reward Estimation* | 72.19 | 4.02 |
| *-w/o Persuasiveness Strategy Optimization* | 75.03 | 4.40 |

to them. Overall, these experimental and human evaluation results suggest that our method not only significantly outperforms open-source community models with similar parameter sizes but also competes effectively with closed-source models that have larger parameter counts.

**Accuracy of Persuasiveness Reward Estimation and Modeling.** Figure 3 shows the Pearson correlation coefficients between various evaluation metrics and human-annotated persuasiveness scores. BLEU-2, ROUGE, and the trained Reward Model all exhibit weak correlations with actual persuasiveness, underscoring the severity of the reward hacking issue. The correlation coefficient between the score ($s$) and true persuasiveness is 0.31, indicating it can reflect persuasiveness to some extent but still contains noise. The separated persuasiveness score ($s_d$) demonstrates high correlation coefficient, while the surface cue score ($s_s$) shows a significantly lower correlation. This illustrates that the Reward Estimation and Modeling accurately isolated the core persuasive elements.

### 4.2 ABLATION STUDY

**Effectiveness of Persuasiveness Reward Estimation and Modeling.** We evaluated the performance without the persuasiveness reward estimation and modeling. Specifically, we used an open-source reward model as the scorer during the persuasiveness strategy optimization phase to provide persuasiveness scores, as shown in Table 2. The experimental results indicate that removing the persuasiveness reward estimation and modeling phase results in a significant decrease of 4.2 points in the o1-score. This suggests that using the existing reward model leads to reward hacking issues, causing performance degradation.

**Effectiveness of Persuasiveness Strategy Optimization.** We assessed the performance without persuasiveness strategy optimization. Specifically, we used the original DPO algorithm for strategy optimization, as shown in Table 2. The experimental results demonstrate that our designed persuasiveness strategy optimization algorithm brought about significant performance improvements, indicating that the enhancements we made to the DPO algorithm for debate and persuasion generation are highly effective.

## 4.3 CASE STUDY

In Appendix C, we present a complete sample including outputs from all baselines and our model. In this example, the original post (OP) was frustrated by the prevalent use of milk bags in Ontario instead of cartons and wanted to be persuaded. As seen, both our model and the closed-source commercial models can provide appropriate arguments and a complete reasoning process. However, our model's arguments and reasoning more directly address the OP's original post, while the outputs from Gemini and Claude contain many generalized or unproven arguments, such as "the prevalence of milk bags in Eastern Canada suggests a successful, albeit different, system established through consumer preference or logistical efficiencies over time", "While milk bags may not be as widely recycled, they generally have a lower environmental impact than cartons", which weaken the persuasiveness of the generated results.

## 5 RELATED WORK

### 5.1 DEBATE AND PERSUASION GENERATION

The task of debate and persuasion generation aims to produce persuasive debate texts for a given topic. With the advancement of large-scale pretrained language models, recent studies often directly leverage LLMs to generate argumentative content. Schiller et al. (2021) proposed a controllable viewpoint generation model capable of generating sentence-level arguments based on a given topic, position, and aspect. Al Khatib et al. (2021) developed three argumentation knowledge graphs and extracted knowledge from them to formulate prompts for training end-to-end viewpoint generation models. Bao et al. (2022) constructed a large-scale argumentative essay generation dataset, ArgEssay. Xiao et al. (2024) introduced the concept of proving principles into LLM planning generation to enhance the persuasiveness of generated texts.

### 5.2 COUNTER ARGUMENT GENERATION

Unlike debate generation, the goal of counter argument generation is to oppose a specific topic or post. Many existing works employ multi-agent frameworks, leveraging conflicts, fusion, and compromises among multiple LLM agents to generate rebuttal sentences (Hu et al., 2023; Xiong et al., 2023; Wang et al., 2023). Other works utilize LLMs' prominent self-reflection capabilities and employ long CoT (Chain of Thought) paradigms to analyze logical flaws in the content to be rebutted, providing stronger counterarguments (Verma et al., 2024; Hu et al., 2023). Some studies are concerned with the impact of using AI tools to aid rebuttal generation on the discussion environment of online debate communities (Zeng et al., 2025).

## 6 LIMITATION

Since our approach relies on crowdsourced annotation data scores, it may not be directly applicable to certain offline debate scenarios. However, thanks to the development of LLM Agent methods (Park et al., 2023), using agent-base user simulators to calculate the number of likes and dislikes for posts and subsequently estimating scores presents a viable alternative solution.

## 7 CONCLUSION

In this paper, we propose a novel training framework, $P^3$, for debate and persuasive text generation tasks to address the shortcomings of LLM training in debating scenarios. This framework focuses on mitigating reward hacking and reward sparsity during model training and optimizes the DPO algorithm's training objectives specific to debate generation. Extensive experiments on the CMV dataset demonstrate that $P^3$ significantly alleviates reward hacking and reward sparsity, substantially improving the persuasiveness of the generated texts. Both automatic evaluation metrics and human assessments show that our method not only surpasses mainstream open-source models but also outperforms state-of-the-art closed-source commercial models such as Gemini and Claude.

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

## A  ALGORITHM FOR PERSUASIVENESS SAMPLE MINING

---

**Algorithm 1** Persuasiveness Sample Mining

---

**Require:** Dataset $D = \{s, s_d\}_{i=1}^{|D|}$, where $s$ represents weak supervised label 'scores' and $s_d$ represents persuasiveness scores for All Posts
**Ensure:** Selected Sequence $L$
 1: Sort the dataset $D$ in descending order by $s_d$ value, and by $s$ value if $s_d$ values are the same
 2: Initialize an empty array $L$ to store the longest sequence
 3: **for** each element $d$ in the sorted dataset $D$ **do**
 4:     Use binary search to find the first element in $L$ that is greater than $d.s$
 5:     **if** such position exists **then**
 6:         Replace the value at that position with $d.s$
 7:     **else**
 8:         Append $d.s$ to the end of $L$
 9:     **end if**
10: **end for**
11: **return** $L$ as the longest sequence

---

## B DATA EXAMPLES AND STATISTICS OF CMV DATASETS

In this section, we present the statistics of the CMV dataset in Table 3, including the number of discussion trees and the number of discussion nodes, among other metrics. Additionally, we provide an example of a discussion tree from the classic CMV dataset in Figure 5.

Table 3: The data statistics for the CMV datasets.

| Type | # Discussion trees | # Nodes | # OPs | # uniq. participants |
|------|--------------------|---------|-------|----------------------|
| Traning | 18,363 | 1,114,533 | 12,351 | 69,965 |
| Test | 2,263 | 145,733 | 1,823 | 16,923 |

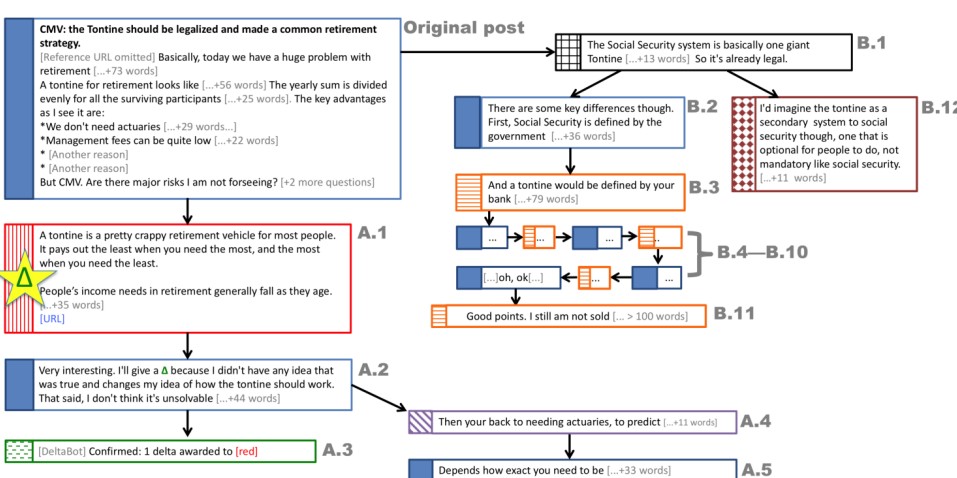

Figure 5: A fragment of a "typical" /r/ChangeMyView discussion tree—typical in the sense that the full discussion tree has an average number of replies (54), although we abbreviate or omit many of them for compactness and readability. Colors indicate distinct users. Of the 17 replies shown (in our terminology, every node except the original post is a reply), the OP explicitly acknowledged only one as having changed their view: the starred reply A.1. The explicit signal is the "Δ" character in reply A.2. The figure is cite from Tan et al. (2016)

## C EXAMPLE OF OUTPUTS

**Original Post**: I moved out to Ontario in 2011 from Alberta, both in Canada, and the first thing I encountered that pissed me off was milk in bags. In Alberta, milk is sold in cartons, and I find it pretty convenient to carry milk around in a fixed-shape container. In Ontario, however, milk almost always comes in a flexible plastic bag. The problems I have with this are:* The shape makes it difficult to store. I have to not only keep unused bags in the fridge, but whatever bag is open needs its own place in the fridge too, AND needs a pitcher to contain it* Cartons are resealable, bags arent́. The number of times that my milk has gone bad since moving here has exploded compared to when I used cartons. The fact that I cant́ re-close my milk bag means it expires faster, and I get less moneyś worth out of it* No other liquids are sold in bags, so why milk? It seems strange that milk is the only drinkable liquid that comes in a bag here. Any other drink thatś in anything resembling a bag, such as Capri Sun or wine, is packaged specifically to optimize consumption (wine in a bag in a box; Capri Sun pouch shaped for drinking), but milk? Itś just in a lazy, unresealable, lame-ass bag.* Bags are bad for the environment. These bags cant́ be re-used, are often not recycled, and I imagine are a huge waste compared to cartons. I know that plastic gallon jugs take more plastic than three bags, but Iv́e always known those jugs to be recycled. Iv́e never seen anyone recycle a milk bag.CMV!

**Gemini**: While the inconvenience of milk bags is understandable, your argument overlooks key economic and environmental factors. The bag system, while less convenient, likely reflects lower transportation and manufacturing costs, resulting in a lower overall price for consumers. Furthermore, the environmental impact isn't solely determined by recyclability; lifecycle assessments comparing bag and carton production, including energy consumption and transportation emissions, might reveal a different overall environmental footprint. Finally, the prevalence of milk bags in Eastern Canada suggests a successful, albeit different, system established through consumer preference or logistical efficiencies over time.

**Claude3**: The blogger's criticism of milk bags in Ontario is understandable, but there are several compelling reasons why the bag system is preferable to cartons: Milk bags are significantly more cost-effective for producers and retailers, allowing them to pass on those savings to consumers. The flexible packaging requires less material and is less expensive to transport, making it a more economical option.The smaller, flexible bags allow for better portion control and reduced waste compared to larger, fixed-size cartons. Consumers can easily pour out the exact amount they need without having to store an open carton. Environmental impact: While milk bags may not be as widely recycled, they generally have a lower environmental impact than cartons. The production and transportation of bags require less energy and resources, and the lighter weight of the packaging reduces fuel consumption.

**Ours**: While it's understandable to feel frustrated about the recycling habits in your area, it's important to recognize that the recyclability of milk bags is a feature of their design, and the issue might lie more with local recycling infrastructure and consumer behavior rather than the product itself. In many regions, including parts of Ontario where milk bags are common, there are established systems for recycling these bags, even if they may not be as visible or widely recognized as those for milk jugs or cartons., the benefits of milk bags extend beyond just being recyclable—they are indeed more efficient in terms of material use and transportation. The reduced weight and volume of milk bags mean fewer resources are needed to produce and transport them, leading to lower overall environmental impact. This includes less fuel consumption during transportation, which translates into a smaller carbon footprint.your concerns about practicality, such as storage and reuse, there are ongoing efforts to improve the user experience. Innovations in refrigerator design, as mentioned, could eventually accommodate milk bags more effectively. Additionally, using a dedicated, reusable, and sealable container for pouring and storing milk from bags is a practical workaround that addresses your specific needs.essence, while the transition to milk bags might require some adjustment, the underlying advantages—both economic and environmental—are significant. By adopting a solution-oriented approach, we can leverage the benefits of milk bags while mitigating any inconveniences.

Table 4: Output example of different baselines and ours.

