# OpenReview forum: "Persuade with Reason: Enhancing Debate Persuasiveness through Accurate Persuasion Feedback Derived from Weak Supervised Labels"
_ICLR.cc/2026/Conference — ICLR 2026 Conference Withdrawn Submission_

### Official Review · Reviewer_nFeL · 2025-10-31

**Soundness:** 3
**Presentation:** 3
**Contribution:** 2
**Rating:** 4
**Confidence:** 3

**Summary:**

The paper addresses the lack of persuasiveness in the debate generation, especially after fine—tuning and RLHF. The main reason for that is reward hacking and reward sparsity. The authors propose P^3, a three-stage pipeline to improve LLM persuasion in debate-style generation: first, separate persuasiveness from surface/literal cues via an EM + Bradley–Terry inspired estimator, then mine persuasive samples from weak supervision (upvote−downvote scores) and filter by the estimated persuasiveness score, and finally, optimize with a modified DPO called PAPO to avoid DPO’s small-sample divergence. Experiments on ChangeMyView (CMV) show improved automatic (GPT-4 o1) and human evaluations; authors claim their 13B model outperforms much larger closed-source models.

**Strengths:**

- The paper is well-structured and easy to follow
- The paper includes both automatic and human evaluations, and shows the correlation between those metrics

**Weaknesses:**

- Lack of a more comprehensive analysis of the generated output. There is only one example described in the case study (section 4.3).
- Unnecessary and redundant equations in section 2 that could have gone to the appendix and free up some space for the analysis of the results (the point above)

**Questions:**

* Why are the human evaluation scores and the o1-score on different scales? Wouldn't it be easier to report the automatic o1-score on the same scale as the human evaluations to facilitate direct comparison?

---

> ### Author Response · Authors · 2025-11-20
>
> ### Thank you for your careful review. We really appreciate your efforts in reviewing our paper.
>
> > Lack of a more comprehensive analysis of the generated output.
>
> In the final version, we will include additional real user examples and analyses. Here, we present a sample output of P3 debating with a human user; this example clearly shows that P3’s persuasiveness primarily arises from stronger logical reasoning and evidence rather than a particular language style.
>
> > Unnecessary and redundant equations in Section 2 that could be moved to the appendix to free up space for result analysis.
>
> Thank you for your suggestion. In the final version, we will adjust the balance between sections and reduce lengthy, low-information equations (for example, the gradient derivations in PAPO will be relocated to the appendix). If you have any questions about the meaning of any formula, please feel free to ask during further discussions.
>
> > Why are the human evaluation scores and the O1-score on different scales?
>
> In fact, the O1-score prompt includes the detailed scoring criteria used in human evaluation. Therefore, the standards for human scoring and the O1-score are indeed the same, despite their different value ranges (O1-score ranges from 0 to 100, while human evaluation ranges from 0 to 5). We will provide detailed human evaluation annotations in the supplementary material.

---

> > ### Author Response · Authors · 2025-11-20
> > **Examples of human debate**
> >
> > User input: I believe that remote work is less productive than working in an office because of the numerous distractions at home and lack of direct supervision. CMV.
> >
> > Model Output: I see where you're coming from, but I think remote work can actually increase productivity. At home, people can tailor their environment to minimize distractions—like choosing a quiet room or using noise-canceling headphones—whereas offices often have open layouts that lead to frequent interruptions. Also, many companies use project management tools to track progress, so direct supervision isn’t always necessary.
> >
> > User Input: While tools exist, I find that without in-person oversight, some employees tend to procrastinate or lose focus. The office environment creates a natural structure and accountability that’s hard to replicate remotely.
> >
> > Model Output: That’s a valid concern, but accountability can be maintained through regular virtual check-ins and clear goal-setting. In fact, studies have found that employees working remotely often report higher job satisfaction and motivation because they have more control over their schedules. This autonomy can lead to better focus and efficiency, rather than less.
> >
> > User: I agree that flexibility is beneficial, but not everyone has the discipline to manage their own time effectively. Plus, some tasks require immediate collaboration, which is easier face-to-face.
> >
> > Model Output: True, but video conferencing and instant messaging tools have significantly improved real-time collaboration. Also, remote work doesn’t mean zero office interaction—it can be hybrid, balancing focused solo work at home with collaborative sessions in person. This model often combines the best of both worlds, enhancing productivity overall.

---

> > ### Author Response · Authors · 2025-11-20
> > **Human annotation standards**
> >
> > The following are detailed criteria for each indicator during manual evaluation.
> >
> > Detailed Standards
> >
> > Score 1: Very Low Persuasiveness
> > The article completely fails to convince me to agree with its main position or arguments.The text presents minimal or weak arguments that barely support the main point, with arguments that are vague, unclear, or poorly structured. It may contain some logical inconsistencies or gaps, making it unlikely to convince the reader.
> >
> > Score 2: Low Persuasiveness
> > Some relevant arguments are provided but they are underdeveloped or superficial. The reasoning is somewhat logical but lacks depth or evidence, and the text may contain minor flaws in coherence or clarity, resulting in limited ability to persuade an attentive reader.
> >
> > Score 3: Moderate Persuasiveness
> > The article partially convinces me to agree with its main position or arguments, but I still maintain an overall disagreement. The text contains clear and relevant arguments supporting the main point, with reasoning that is generally logical and mostly coherent. Some evidence or examples may be provided but are not strong or comprehensive, allowing the text to moderately influence the reader’s opinion.
> >
> > Score 4: High Persuasiveness
> > The article overall convinces me to agree with its main position or arguments, but I still have some doubts about certain details. Arguments are well-developed, relevant, and logically structured, providing convincing evidence or examples to support claims. The text is clear, coherent, and engaging, making it likely to persuade most readers.
> >
> > Score 5: Extremely Persuasive
> > The article completely convinces me to agree with its main position or arguments. The text presents compelling, comprehensive, and well-articulated arguments, using strong evidence, examples, and reasoning that leave little room for doubt. It is highly coherent, eloquent, and emotionally or logically engaging, almost certainly convincing the reader of the intended viewpoint.

---

### Official Review · Reviewer_s2Y1 · 2025-11-01

**Soundness:** 2
**Presentation:** 3
**Contribution:** 2
**Rating:** 4
**Confidence:** 4

**Summary:**

This paper proposes PAPO, a persuasion-aware preference optimization framework. Using the CMV dataset, the authors disentangle persuasiveness from style by modeling two latent components and optimizing through a score-weighted objective that addresses divergence issues in standard DPO. The model is evaluated both automatically and with limited human assessments, showing improved alignment with human judgments compared to existing preference optimization methods.

**Strengths:**

The study addresses a gap between language fluency and genuine persuasiveness in LLM outputs. Its attempt to separate logical argument quality is conceptually novel. The integration of theoretical justification and practical large-scale experiments make the paper methodologically grounded.

**Weaknesses:**

Comment 1. The authors fit two MLPs (sd for persuasion, ss for surface) and learns a Bernoulli-mixture with EM. It’s unclear what sd truly captures. Does sd represent causal persuasiveness (logic, evidence) or merely represent latent artifacts?
Using a small subsample, the authors might test the following.
(i) hold content constant while changing style; (ii) hold style constant while changing content. Check that sd is stable in (i) and changes in (ii); ss should do the opposite.

Comment 2. The Reddit-based supervision signal (upvotes minus downvotes), as the authors note, can conflate persuasiveness with unrelated noise, such as popularity and timing effects. This weak supervision may cause the model to reward factors other than genuine argumentative strength. This is particularly important as the entire algorithm hinges on this “weak” supervision.
The authors can probably include fixed effects for subreddit, posting time, and author karma, and then re-estimate the learned persuasion scores to test robustness.
Furthermore, the authors can use CMV “delta” awards as clean persuasion labels, train on that subset, and compare whether PAPO still outperforms the traditional DPO in that subset.

Comment 3. I am not particularly convinced by the reliability of o1 scores. The authors’ argument is based on limited validation (only 100 human-rated samples).
Potentially, the authors can consider increasing the human sampling stratified by topic, stance, and argument length. Importantly, it will be interesting to re-prompt the evaluator with alternative rubrics (logic-only vs. style-only) and show that model rankings remain stable.

Comment 4. The CMV dataset is known to have topic imbalances. To ensure that the authors’ results can generalize across different topics, the authors can split CMV data by topic clusters and show whether improvements hold uniformly or only in high-frequency topics. It will be worthwhile to investigate under which conditions (by topic or by any other systematic characteristics) the algorithm performs relatively better or worse.

Comment 5. The title of paper is currently missing in your submission.

**Questions:**

See the above "Weaknesses."

---

> ### Author Response · Authors · 2025-11-20
>
> ### Thank you for your careful review. We really appreciate your efforts in reviewing our paper.
>
> > Comment 1: The Meaning of S_s and S_d
>
> In our designed EM algorithm, the main difference between S_s and S_d lies in their intended focus. S_s is designed to capture surface textual cues, such as writing style and expression. In contrast, S_d aims to capture the true persuasiveness of the text, such as causal and logical relationships. We assume that surface cues related to wording are contained solely within the input x, whereas true persuasiveness requires considering the interaction between x, the reply history, and the original post by the OP. Accordingly, in the functional formulation, S_s takes only the text x itself as input, while S_d incorporates the interaction between the current post and the reply history. As shown in Figure 3, we observe that S_d correlates significantly higher with true persuasiveness than S_s, indicating that the EM algorithm successfully captures genuine persuasive information. Naturally, as you pointed out, since these are estimated via latent variables, we cannot precisely know how S_d and S_s are evaluated. Therefore, following your suggestion, we conducted supplementary experiments. The results show that when style changes but content remains constant, S_d remains largely stable while S_s varies significantly, confirming the accuracy of our estimation method. We will include this experiment in the final version of the paper.
>
> > Comment 2: Noise in Score Labels
>
> As you noted, the Score labels include many factors unrelated to persuasiveness, such as writing style and topic popularity. Regarding surface cues like writing style, we use the EM algorithm to disentangle true persuasiveness from superficial information, as explained in the rebuttal above. To address topic popularity, we normalize score values within each discussion tree to facilitate comparison across topics. Additionally, when selecting positive and negative samples for DPO training, we only compare persuasiveness scores within the same discussion tree. This approach minimizes the impact of popularity differences across topics and themes on the final results. We also conducted an additional experiment using delta labels for PAPO training. Due to the scarcity of delta labels, models trained this way do not surpass the performance of closed-source commercial models but do outperform the untrained base model, demonstrating the effectiveness of our method.
>
> > Comment 3: Concerns about the Reliability of o1 Scores
>
> We analyzed the accuracy and reliability of o1 scores, finding that their correlation with human evaluations is within an acceptable range (0.67). While this correlation is not perfect, given the enormous scale of the CMV dataset (which includes 1,114,533 discussion nodes), the o1 score remains the only viable automatic evaluation method. In the final evaluation, we also plan to add more human annotations and, as you suggested, conduct detailed analyses by topic and hierarchical level. We also plan to incorporate additional human evaluation experiments in the final version of the paper, such as debate sessions with real human users. We will include a sample output of P3 debating with a human user; from this example, it is evident that P3’s persuasiveness primarily stems from stronger logical reasoning and evidence rather than specific language style.
>
> > Comment 4: Topic Imbalances in the CMV Dataset
>
> As we addressed in Comment 2, P3 has already considered and attempted to mitigate the variance in persuasiveness reward estimation caused by heterogeneity across different topics. However, as you pointed out, we can further enhance our analysis. We will include clustering of CMV topics and conduct topic-wise analyses of experimental results in the final version of the paper.
>
> > Comment 5. The title of paper is currently missing in your submission.
>
> We will correct this error in the final version.

---

> > ### Author Response · Authors · 2025-11-20
> > **Delta Label Training**
> >
> > In this response, we provide the results of applying PAPO using delta labels.
> >
> > | Model | o1-score | human-eval |
> > |------|------------|---------------|
> > | Qwen2-13B | 73.57 | 4.10 |
> > | Qwen2-72B | 75.73 | 4.32 |
> > | Gemini1.5 Flash | 77.01 | 4.60 |
> > | Ours | **76.02** | **4.36** |

---

> > ### Author Response · Authors · 2025-11-20
> > **Examples of human debate**
> >
> > User input: I believe that remote work is less productive than working in an office because of the numerous distractions at home and lack of direct supervision. CMV.
> >
> > Model Output: I see where you're coming from, but I think remote work can actually increase productivity. At home, people can tailor their environment to minimize distractions—like choosing a quiet room or using noise-canceling headphones—whereas offices often have open layouts that lead to frequent interruptions. Also, many companies use project management tools to track progress, so direct supervision isn’t always necessary.
> >
> > User Input: While tools exist, I find that without in-person oversight, some employees tend to procrastinate or lose focus. The office environment creates a natural structure and accountability that’s hard to replicate remotely.
> >
> > Model Output: That’s a valid concern, but accountability can be maintained through regular virtual check-ins and clear goal-setting. In fact, studies have found that employees working remotely often report higher job satisfaction and motivation because they have more control over their schedules. This autonomy can lead to better focus and efficiency, rather than less.
> >
> > User: I agree that flexibility is beneficial, but not everyone has the discipline to manage their own time effectively. Plus, some tasks require immediate collaboration, which is easier face-to-face.
> >
> > Model Output: True, but video conferencing and instant messaging tools have significantly improved real-time collaboration. Also, remote work doesn’t mean zero office interaction—it can be hybrid, balancing focused solo work at home with collaborative sessions in person. This model often combines the best of both worlds, enhancing productivity overall.

---

### Official Review · Reviewer_7znS · 2025-11-02

**Soundness:** 2
**Presentation:** 2
**Contribution:** 2
**Rating:** 4
**Confidence:** 2

**Summary:**

This paper tackles the challenge of generating persuasive debate text with large language models. The authors identify two major issues that cause existing methods, especially supervised fine-tuning (SFT) and reinforcement learning with human feedback (RLHF), to fail in producing truly persuasive arguments. To address these, the authors propose P3, a three-stage training framework.

**Strengths:**

Clear Problem Identification: The paper convincingly argues that traditional RLHF pipelines are poorly aligned for debate/persuasion tasks due to reward misalignment (hacking) and data sparsity.

Well-Motivated Framework: The P3 pipeline logically decomposes the overall goal into reward estimation, data selection, and strategy optimization. Each component addresses a specific, well-defined issue.

**Weaknesses:**

1. Limited Dataset Scope: All experiments are restricted to the CMV dataset. While appropriate for persuasion, results on other argumentative or dialogue datasets (e.g., ConvAI, PersuasionForGood) would demonstrate generalization.

2. Using GPT-4-o1 as the primary automatic evaluator introduces circular dependence, since GPT-4’s reward alignment may favor stylistic fluency over genuine logical rigor. The correlation coefficient (0.67) is reasonable but not perfect; more robust human-only evaluations would strengthen the conclusion.

**Questions:**

Please refer to the weakness part.

---

> ### Author Response · Authors · 2025-11-20
>
> ### Thank you for your careful review. We really appreciate your efforts in reviewing our paper.
>
> > Limited Dataset Scope
>
> We chose the CMV dataset as our primary evaluation benchmark mainly because it is currently the largest dataset in terms of volume, covers the widest range of topics, and contains fine-grained “persuasion” labels. The full CMV dataset includes 1,114,533 discussion nodes (analogous to dialogue turns) and 18,363 discussion trees (analogous to topics).  Nonetheless, testing on these online debate datasets is still necessary to demonstrate the robustness and generalization of our approach. Therefore, we have conducted supplementary experiments on these datasets; due to space constraints, detailed results are included in the supplementary responses. The experimental results show that our method significantly outperforms strong baselines in out-of-domain online debate scenarios, confirming its effectiveness, robustness, and generalization ability. This also supports the representativeness of the CMV benchmark.
>
> > Using GPT-4-o1 as the primary automatic evaluator introduces circular dependence, since GPT-4’s reward alignment may favor stylistic fluency over genuine logical rigor. The correlation coefficient (0.67) is reasonable but not perfect; more robust human-only evaluations would strengthen the conclusion.
>
> One major reason we selected o1 as the evaluation model is that its technical report indicates it was trained and evaluated specifically for persuasiveness [1]. Naturally, as you noted, the correlation coefficient of 0.67 is not perfect. We also plan to incorporate additional human evaluation experiments in the final version of the paper, such as debate sessions with real human users. We will include a sample output of P3 debating with a human user; from this example, it is evident that P3’s persuasiveness primarily stems from stronger logical reasoning and evidence rather than specific language style.
>
> [1] Openai o1 system card

---

> > ### Author Response · Authors · 2025-11-20
> > **Online debate supplementary experiment**
> >
> > In this response, we will provide the results on out-domain online debate dataset DDO corpus (mentioned in [1] and [2]). Due to the time limitations, we only select 100 test samples and three strong baselines. We will add full results in the final version of the paper.
> >
> > [1] Exploring the Role of Prior Beliefs for Argument Persuasion
> > [2] A Corpus for Modeling User and Language Effects in Argumentation on Online Debating
> >
> > | Model | o1-score | human-eval |
> > |------|------------|---------------|
> > | Qwen2-13B | 75.62 | 4.30 |
> > | Qwen2-72B | 77.35 | 4.42 |
> > | Gemini1.5 Flash | 80.21 | 4.66 |
> > | Ours | **83.32** | **4.73** |

---

### Official Review · Reviewer_mPUe · 2025-11-02

**Soundness:** 2
**Presentation:** 2
**Contribution:** 2
**Rating:** 2
**Confidence:** 3

**Summary:**

The paper focus on persuasion generation, creating text that convinces a specific audience. The authors attribute the limitation of current persuasion generation to 2 main reasons: Reward hacking (objective design) and Reward sparsity (training data).
To tackle the deconstructed issues, the enhance pipeline, $P^3$, is proposed. The pipeline scores generations along literal and persuasiveness dimensions and trains using weakly supervised labels. Experiments on CMV demonstrate its effectiveness in improving persuasive quality.

**Strengths:**

1. Problem decomposition: deconstructing the weaknesses of persuasion generation along two axes, objective design and data scarcity, to diagnose failures and guide targeted remedies.
2. Fine-grained scoring: decoupling the reward into literal fidelity and persuasiveness, enabling fine-grained optimization of persuasive quality.
3. Data construction pipeline: propose a scalable data-construction pipeline that alleviates data scarcity and provides a practical recipe for weak/cheap supervision.

**Weaknesses:**

1. Fig. 1 inconsistency (objective vs. paradigm).
The paper attributes failures to objective design, but Fig. 1 varies training paradigms (SFT vs. RL) rather than the objective itself. As a result, the figure does not isolate objective mis-specification.

2. Limited evaluation scope (single dataset).
The primary evaluation is on CMV; broader experiments on additional persuasion/debate datasets would strengthen external validity and generality of the conclusions.

3. Baseline parity (backbone vs. fine-tuned).
In Tab. 1 and Fig. 4, comparisons appear to pit the proposed fine-tuned system against general-purpose backbones. For a fair test of $P^3$’s contribution, including equally fine-tuned baselines can consolidate the final conclusion

4. Strong model comparison.
Given the reported high correlation between o1-score and human annotations, GPT-4o seems to be a strong baseline models which is missing in the performance comparison.

5. Missing specific metric was used for human evaluation.
The paper lacks a clear description or example of the human scoring process, making the evaluation criteria ambiguous. Providing concrete examples or a defined scoring rubric would clarify how the human judgments were obtained and interpreted.

6. The title in the manuscript is inconsistent with the title provided in the submission page.

**Questions:**

1. In Fig.2, more detailed statement is required. Why do such scatter plots demonstrate the deviation?
2. In formula 3, what is the $x$ in $f_s(x)$ standing for
3. same one in formula 5
4. in Fig.3, given that the o1-score shows the highest correlation with human judgments, would a simple pipeline that uses GPT-4o1 to automatically annotate data and then trains SFT/RL models without any task-specific tailoring yield promising gains on persuasion generation?
5. For automatic evaluations, what are the prompt templates (scoring rubric, temperature, top-p, max tokens).

---

> ### Author Response · Authors · 2025-11-20
>
> ### Thank you for your careful review. We really appreciate your efforts in reviewing our paper.
>
> > Fig. 1 inconsistency (objective vs. paradigm).
>
> Figure 1 primarily illustrates the phenomenon that both SFT and RL lead to a decrease, rather than an increase, in model persuasiveness on the persuasion task. We attribute this to a misalignment between the training objectives and persuasiveness. Although different training methods are presented, applying various alignment strategies to the same training objective consistently results in performance degradation compared to the pre-training baseline. This indicates that the training and alignment objectives themselves do not accurately capture textual persuasiveness. This issue is further reflected in Figure 3, where word-overlap-based metrics such as BLEU and ROUGE show very low correlation with persuasiveness.
>
> > Limited evaluation scope (single dataset).
>
> We chose the CMV dataset as our primary evaluation benchmark mainly because it is currently the largest dataset in terms of volume, covers the widest range of topics, and contains fine-grained “persuasion” labels. The full CMV dataset includes 1,114,533 discussion nodes (analogous to dialogue turns) and 18,363 discussion trees (analogous to topics).  Nonetheless, testing on these online debate datasets is still necessary to demonstrate the robustness and generalization of our approach. Therefore, we have conducted supplementary experiments on these datasets; due to space constraints, detailed results are included in the supplementary responses. The experimental results show that our method significantly outperforms strong baselines in out-of-domain online debate scenarios, confirming its effectiveness, robustness, and generalization ability. This also supports the representativeness of the CMV benchmark.
>
> > Baseline parity (backbone vs. fine-tuned).
>
> It should be noted that the closed-source commercial models involved in this study have also been trained on CMV; for example, the GPT-4o1 technical report explicitly includes training and evaluation on the CMV task [1]. Some models, such as Claude, have been specifically optimized for persuasion-related tasks [2], making our comparisons fair. Naturally, as you suggested, including additional baselines trained on this task helps better demonstrate the practical performance of our method. Therefore, we added the PESA baseline [3], whose training objective also focuses on enhancing model persuasiveness. Experimental results show that our approach significantly outperforms this strong baseline on persuasion metrics. Due to space limitations, we will include these results in the supplementary response.
>
> [1] Openai o1 system card
>
> [2] Measuring the Persuasiveness of Language Models: https://www.anthropic.com/news/measuring-model-persuasiveness
>
> [3] Prove Your Point!: Bringing Proof-Enhancement Principles to Argumentative Essay Generation
>
> > Strong model comparison.
>
> It is important to note that both Claude and Gemini are powerful closed-source commercial models. We did not use GPT-4o because the evaluation employed the O1 model, and we were concerned about potential bias favoring GPT-4o within the same model family. To address your concern, we compared the outputs generated by GPT-4o; the experiments show that texts generated by P3 exhibit persuasiveness superior to those of GPT-4o.
>
> > Missing specific metric was used for human evaluation.
>
> We will provide detailed criteria for our human evaluation in the final version of the paper, as well as in the supplementary response for this review. Regarding annotators' stances and beliefs, we require all annotators to maintain a neutral position on the annotation tasks, without favoring any viewpoint. If an annotator finds that they hold strong preconceived biases on a particular task, they must notify us, and we will replace them accordingly. Consequently, the three annotators who completed the labeling maintained neutrality on the annotated issues, ensuring that their stances did not affect the accuracy of the persuasiveness annotations.
>
> > The title in the manuscript is inconsistent with the title provided in the submission page.
>
> We will correct this error in the final version.

---

> > ### Author Response · Authors · 2025-11-20
> > **Online debate supplementary experiment**
> >
> > In this response, we will provide the results on out-domain online debate dataset DDO corpus (mentioned in [1] and [2]). Due to the time limitations, we only select 100 test samples and three strong baselines. We will add full results in the final version of the paper.
> >
> > [1] Exploring the Role of Prior Beliefs for Argument Persuasion
> >
> > [2] A Corpus for Modeling User and Language Effects in Argumentation on Online Debating
> >
> > | Model | o1-score | human-eval |
> > |------|------------|---------------|
> > | Qwen2-13B | 75.62 | 4.30 |
> > | Qwen2-72B | 77.35 | 4.42 |
> > | Gemini1.5 Flash | 80.21 | 4.66 |
> > | Ours | **83.32** | **4.73** |

---

> > ### Author Response · Authors · 2025-11-20
> > **SFT baselines**
> >
> > In this response, we will provide the results compared to PESA baseline.
> >
> > | Model | o1-score | human-eval |
> > |------|------------|---------------|
> > | PESA | 74.50 | 4.20 |
> > | Ours | **78.29** | **4.62** |

---

> > ### Author Response · Authors · 2025-11-20
> > **GPT4-o baseline supplementary experiment**
> >
> > In this response, we will provide the results compared to GPT4-o.
> >
> > | Model | o1-score | human-eval |
> > |------|------------|---------------|
> > | Qwen2-13B | 73.57 | 4.10 |
> > | Qwen2-72B | 75.73 | 4.32 |
> > | Gemini1.5 Flash | 77.01 | 4.60 |
> > | Claude3 Haiku | 77.85 | 4.58 |
> > | GPT4-o | 77.52 | 4.54 |
> > | Ours | **78.29** | **4.62** |

---

> > ### Author Response · Authors · 2025-11-20
> > **Human annotation standards**
> >
> > The following are detailed criteria for each indicator during manual evaluation.
> >
> > Detailed Standards
> >
> > Score 1: Very Low Persuasiveness
> > The article completely fails to convince me to agree with its main position or arguments.The text presents minimal or weak arguments that barely support the main point, with arguments that are vague, unclear, or poorly structured. It may contain some logical inconsistencies or gaps, making it unlikely to convince the reader.
> >
> > Score 2: Low Persuasiveness
> > Some relevant arguments are provided but they are underdeveloped or superficial. The reasoning is somewhat logical but lacks depth or evidence, and the text may contain minor flaws in coherence or clarity, resulting in limited ability to persuade an attentive reader.
> >
> > Score 3: Moderate Persuasiveness
> > The article partially convinces me to agree with its main position or arguments, but I still maintain an overall disagreement. The text contains clear and relevant arguments supporting the main point, with reasoning that is generally logical and mostly coherent. Some evidence or examples may be provided but are not strong or comprehensive, allowing the text to moderately influence the reader’s opinion.
> >
> > Score 4: High Persuasiveness
> > The article overall convinces me to agree with its main position or arguments, but I still have some doubts about certain details. Arguments are well-developed, relevant, and logically structured, providing convincing evidence or examples to support claims. The text is clear, coherent, and engaging, making it likely to persuade most readers.
> >
> > Score 5: Extremely Persuasive
> > The article completely convinces me to agree with its main position or arguments. The text presents compelling, comprehensive, and well-articulated arguments, using strong evidence, examples, and reasoning that leave little room for doubt. It is highly coherent, eloquent, and emotionally or logically engaging, almost certainly convincing the reader of the intended viewpoint.

---

> ### Author Response · Authors · 2025-11-20
>
> > In Fig.2, more detailed statement is required.
>
> In the scatter plot of Figure 2, the x-axis represents the values of the respective metrics indicated in the title (e.g., BLEU scores for the BLEU plot), while the y-axis represents the O1-score, which reflects textual persuasiveness. This figure primarily highlights the limitations of BLEU and ROUGE in evaluating persuasiveness; that is, whether based on word-overlap metrics like BLEU/ROUGE or on a trained Reward Model, their correlation with actual persuasiveness remains low.
>
> > Notation in formula
>
> The input x in formulas (3) and (5) represents the Reddit post to be analyzed; for example, f_s(x) denotes the surface clue score of post x, which includes information such as language style unrelated to persuasiveness.
>
> > Would a simple pipeline that uses GPT-4o1 to automatically annotate data and then trains SFT/RL models without any task-specific tailoring yield promising gains on persuasion generation
>
> We believe this helps mitigate reward hacking and the issue of sparse rewards. However, as noted in the rebuttal above, the full CMV dataset contains 1,114,533 discussion nodes (analogous to dialogue turns) and 18,363 discussion trees (analogous to topics). The time and financial costs of annotating all of them using the o1 model are prohibitive. Therefore, we adopt a more sophisticated approach to extract persuasiveness scores from weakly supervised labels. As shown in Figure 3, the persuasiveness scores S_d we extract also exhibit a strong correlation with human evaluations.
>
> > Automatic evaluations settings
>
> The automatic evaluation prompt uses the same standard wording as the human evaluation and requires the model to generate a persuasiveness score ranging from 0 to 100. For other parameters, we set temperature = 0, top-p = 1, and max tokens = 1024.

---

> > ### Comment · Reviewer_mPUe · 2025-11-24
> >
> > > In Fig. 2(a), more detailed explanation is needed.
> > 1. I am still confused about Fig.2. Why O1-score could represents human annotators (as stated in **L064**)
> > 2. what is the data used in Fig.2(a), without the scale in the x-, y-axis, the visualization will be less convincing
> > 3. could the authors describe the the case where metrics does not diverge from the persuasiveness scores given by human annotators should look like. Such comparison could more directly demonstrate
> >
> > > Notation in formula
> >
> > Could you further clarify the multiple notation in paper:
> > $x$: Reddit post;
> > $x_p$: original post (L144).
> > **What is the difference or relationship between them?**
> >
> > > Would a simple pipeline that uses GPT-4o1 to automatically annotate data and then trains SFT/RL models without any task-specific tailoring yield promising gains on persuasion generation
> >
> > Thanks for the further clarification. The proposed persuasiveness reward estimation and persuasiveness strategy optimization are indeed impressive and show the authors’ deep consideration of this problem. However, my main concern is that the reliance on crowd-sourced annotation scores (from the CMV dataset) substantially narrows the practical scope of the paper. As noted in the Limitations section, LLM agents are still required to improve generalization in certain offline debate scenarios. In that case, if we already need additional LLM agents, how do the authors justify the efficiency of the proposed customized pipeline compared to simply using GPT-4(o) for annotation and then training SFT/RL models on those labels?
> >
> > In my understanding, the main issue in current persuasion generation is the lack of fine grained annotations that provide effective guidance for learning. If the authors could demonstrate that the pipeline described in the Method section is necessary or clearly advantageous for persuasion generation, compared with simply collecting or generating more fine grained annotations and training with standard SFT or RL, it would greatly strengthen the contribution of the paper.

---

> ### Comment · Reviewer_mPUe · 2025-11-24
>
> I appreciate the further clarification and the extra effort the authors invest.
>
> > Fig. 1 inconsistency (objective vs. paradigm).
>
> First, thanks the further clarification. I understand the authors try to use Fig.1 the point out the limitation of current objective design. However, my concern in here is the authors negate the large scope (training paradigms which could use multiple customized objectives) first, although the main issue discussed in paper is fall in the another scope (training objective).
> Like the  statement begin at **L055**, it is too strong to negate SFT and RL without specifying the certain training objective. Like statement in line071, the key point is the training objective, rather than the training paradigm.
> The alternative framing could be: with an inappropriate objective, existing training paradigms such as SFT and RL can even degrade performance in persuasiveness generation after post-training.
> In contrast, the current version, initially, demonstrate the problem to the training paradigm itself, which can be misleading and misaligned with the rest of the paper, where a new objective is actually used within an existing paradigm (RL).
>
> > Limited evaluation scope (single dataset).
>
> Additional experiments would indeed improve the generalizability of this work, and such results should be added in the revised version to further strengthen the paper’s contribution.

---

> ### Comment · Reviewer_mPUe · 2025-11-24
> **More questions and concerns**
>
> After carefully reviewing the submitted paper, I have encountered some additional points of confusion. I would appreciate it if the authors could further clarify these, as it would help me better understand the work.
>
> > question about Eq.3
>
> About $S = f_s(x) - f_s(x_p)$, could you explain more about meaning or purpose of this equation. What is the purpose or meaning of `-` in here?
>
> > ablation study in $\alpha$ in Eq.3
>
> $\alpha$ is a hyperparameter in the **persuasiveness modeling**, how do you determine the value of it in the main experiment? Does varying $\alpha$ have a significant impact on the final performance?
> Does the optimal $\alpha$ depend on the training datasets? If so, does a different optimal $\alpha$ across datasets indicate something meaningful, and could you provide some insight into this?
>
> ---
> The following points are about presentation issues:
>
> > the definition of OP
>
> The first appearance of OP is at **L145**, but its definition appears later at L309 (“… an original poster (OP) shares …”). Moving this definition earlier in the paper would improve readability.
>
> > Mistake in **L217**?
>
> In **L217**: " ... hidden variables $S_s$ (**persuasiveness score**) and $S_d$ (**literal score**)." Is it a mistake?

---

> > ### Author Response · Authors · 2025-11-24
> >
> > Thank you for your thoughtful and constructive response. I will address your concerns in hopes of resolving them.
> >
> > > it is too strong to negate SFT and RL without specifying the certain training objective
> >
> > Indeed, what we intended to convey in the P3 paper is that, for the specific task of persuasiveness, current training objectives—whether based on word and text similarity or on post-training reward models—are biased to the extent that models fine-tuned with SFT or RL actually degrade in performance. To address this, we identified and tackled two issues: reward hacking and reward sparsity. In the original introduction, we stated, “Additionally, whether fine-tuning large language models (LLMs) directly or using reinforcement learning with human feedback (RLHF), the persuasiveness of the generated text tends to decline, making it challenging to leverage advancements in state-of-the-art LLM technologies for this task.” To reduce potential ambiguity in this statement, we will clarify in the final version the distinction between training objectives and training paradigms.
> >
> > > Limited evaluation scope (single dataset).
> >
> > We fully agree with your point. Therefore, we incorporated an additional, larger, and widely used dataset in debate generation, DDO, and conducted supplementary experiments that yielded quite positive results. Please refer to the “Online debate supplementary experiment” in our earlier reply for details. We will include the full results of this experiment in the final paper.
> >
> > > In Fig. 2(a), more detailed explanation is needed.
> >
> > 1) There was an error in the legend labeling: legend (a) should be corrected to “scatter plot of metrics and automatic persuasion metrics.” Regarding whether the O1-score represents human annotation, we discussed this in the Results and Analysis section. Figure 3 shows the Pearson correlation between O1-score and human-rated persuasiveness, which is relatively high (0.67) and exceeds that of other automatic metrics such as BLEU and ROUGE.
> >
> > 2) Figure 2 uses the CMV dataset, from which we randomly sampled 50 instances. The x-axis shows scores from BLEU, ROUGE, and the reward model for each sample, while the y-axis shows the O1-score.
> >
> > 3) Ideally, an automated persuasiveness metric should be perfectly linearly correlated with true persuasiveness; thus, the scatter plot would approximate a line extending from the bottom right to the top left.
> >
> > > Notation in formula
> >
> > For $ x_p $, it denotes the original post (OP) in the CMV dataset. In this context, the OP typically presents their viewpoint and invites others to “Change My View,” making them the target user our model aims to persuade.
> >
> > For $ x $, it represents posts by other users within the CMV discussion tree. These users may attempt to persuade the OP or counter each other, forming a complex, multi-turn, multi-party debate environment.
> >
> > > Would a simple pipeline that uses GPT-4o1 to automatically annotate data and then trains SFT/RL models without any task-specific tailoring yield promising gains on persuasion generation?
> >
> > As explained in our previous response, the time and financial costs of using GPT-4o1 to annotate the entire CMV dataset are prohibitive. For small batches of data or simply collected labels, we included a supplementary Delta Label Training experiment in our reply to reviewer s2Y1. Specifically, the CMV dataset contains a small number of fine-grained persuasiveness labels: some OPs mark delta labels on posts within the discussion tree to acknowledge being persuaded. Although these labels are more precise and accurate than GPT-4o1 annotations, their quantity is extremely limited (about 1%), leading to reward sparsity and suboptimal performance. Our experiments show that models trained with these delta labels do not surpass the performance of our method, which validates the effectiveness of our approach.
> >
> > On the other hand, supplementary experiments in our response also demonstrate that although models trained on CMV do not fully generalize to some online debate data, they still outperform closed-source commercial models on the DDO dataset, indicating good generalizability and applicability of our method.

---

> > > ### Comment · Reviewer_mPUe · 2025-11-26
> > >
> > > Appreciating the response, which makes the main idea more clear to me
> > >
> > > > Presentation-related
> > > 1. `strong negation in SFT and RL`: after revised, it could largely solve my concern.
> > > 2. `Fig. 2(a) Issue`: after authors revise the paper in such part, it will make it more clear, and clear my original concern about it
> > > 3. `Notation $x$ in formula`: the rely about it is largely helpful, considering adding introduction about it in the main paper or appendix (since there is not official definition about $x$ before using it), and the revised version solve my concern.
> > >
> > > > Would a simple pipeline that uses GPT-4o1 to automatically annotate data and then trains SFT/RL models without any task-specific tailoring yield promising gains on persuasion generation?
> > >
> > > Fully understand the constraints imposed by time and financial costs and appreciate the effort demonstrated in [Delta Label Training](https://openreview.net/forum?id=lmQjnEIAv6&noteId=rJd2AA3R1E).
> > >
> > > While I apologize for any previous lack of clarity, my main concern lies in the **comparison between training SFT/RL models with fine-grained annotations** (e.g., Delta Labels) **and the proposed customized pipeline** (assuming the absence of such annotations), **under the same scale of training data**. Rather than PAPO using delta labels.
> > >
> > > For a quick justification, using the full CMV dataset is not necessary. Instead, at the **Delta Label scale**, a question is:
> > > 1. Does the proposed pipeline outperform the model trained via SFT/RL with Delta Labels? or
> > >
> > > 2. Does the proposed pipeline outperform the model trained via SFT/RL with GPT-annotated data? (If this comparison is unavailable, the first one would already provide sufficient justification.)
> > >
> > > Such a comparison could serve as an effective and efficient way to demonstrate the necessity and advantage of the proposed pipeline.

---

> > ### Author Response · Authors · 2025-11-24
> >
> > > Question about Eq. 3
> >
> > In the formula $ S = f_s(x) - f_s(x_p) $, $ f_s(x) $ denotes the surface cue score of $ x $, and $ f_s(x_p) $ denotes that of $ x_p $. Their difference represents the extent to which the debater’s writing style, rhetorical skills, and other surface-level features surpass those of the OP. Our main assumption is that surface cues relate solely to the text itself; thus, the function $ f_s $ evaluates only the post independently, whereas the persuasiveness score $ f_d $ considers the entire dialogue history, including logical interactions between the post and the OP. Since $ f_s $ assesses only the post itself and cannot directly express how much it surpasses the OP, we subtract the OP’s $ f_s $ score to capture this relative advantage.
> >
> > > Ablation study on Eq. 3
> >
> > We performed additional ablation experiments on the parameter $\alpha$, which reveal that the O1-score follows a unimodal pattern with optimal performance near $\alpha = 0.6$. This suggests that in actual Reddit discussions, only a minority of the audience evaluates persuasiveness based on textual content, while the majority rely more on superficial features such as writing style.
> >
> > > The following points are about presentation issues
> >
> > Thank you for pointing out these errors. We sincerely apologize for these issues and will address them in the final version of the paper.

---

> > ### Author Response · Authors · 2025-11-24
> > **Delta Label Training**
> >
> > In this response, we provide the results of applying PAPO using delta labels.
> >
> > | Model | o1-score | human-eval |
> > |------|------------|---------------|
> > | Qwen2-13B | 73.57 | 4.10 |
> > | Qwen2-72B | 75.73 | 4.32 |
> > | Gemini1.5 Flash | 77.01 | 4.60 |
> > | Ours | **76.02** | **4.36** |

---

> > ### Author Response · Authors · 2025-11-24
> > **Supplementary Hyperparameter Experiment**
> >
> > In this response, we present the effect of the hyperparameter alpha on the O1-score.
> >
> > | Alpha | o1-score |
> > |------|------------|
> > | 0.4 | 70.23 |
> > | 0.5 | 75.62 |
> > | 0.6 | **78.29** |
> > | 0.7 | 77.20 |
> > | 0.8 | 75.01 |

---

> ### Comment · Reviewer_mPUe · 2025-11-26
>
> Thanks for the further clarification, which solves most of my confusions and question.
>
> After revising all the issues mentioned previous, I believe the paper deserves a better score, and I will raise my score to 4.
>
> However, currently, my main concern still in *Would a simple pipeline that uses GPT-4o1 to automatically annotate data and then trains SFT/RL models without any task-specific tailoring yield promising gains on persuasion generation?*, as shown in this [reply](https://openreview.net/forum?id=lmQjnEIAv6&noteId=BFHP8J1Rbx).

---

### Official Review · Reviewer_ChvZ · 2025-11-03

**Soundness:** 2
**Presentation:** 1
**Contribution:** 2
**Rating:** 2
**Confidence:** 4

**Summary:**

The paper presents a persuasiveness-enhanced preference optimization framework that extends DPO to model persuasive reasoning under weak supervision. It leverages social feedback signals to learn from large-scale debate data and employs a dual-reward mechanism to separate genuine persuasion from superficial style. Experiments on persuasion and debate benchmarks show consistent gains in both automatic and human evaluations. Overall, the approach provides a simple yet effective way to align language models toward persuasive and content-driven generation under weakly supervised settings.

**Strengths:**

1.The paper introduces a novel persuasiveness-aware extension of DPO that separates logical persuasion from surface-level style through dual reward modeling.
2. The paper effectively leverages large-scale social feedback as weak supervision, demonstrating how noisy real-world data can be adapted for persuasive language modeling.

**Weaknesses:**

1. The paper’s exposition is often unclear and difficult to understand.
2. The experiments compare P³ mainly with general instruction-tuned LLMs (e.g., LLaMA, Claude, Gemini), but omit specialized baselines that also target persuasion or argumentation, such as: SFT-based methods, DPO or similar methods are focusing on reward hacking and reward sparsity.
3. The paper reports only the O₁-score (GPT-4 evaluation) and limited human judgments. Even though some previous metrics have limitation, it can reflect something by using Bert score, Rouge/BLEU. Such as semantic alignment and style or phrasing pattern.
4. The paper lacks both statistical and qualitative analyses of generated outputs. There is no examination of response length, lexical diversity, or argument structure, nor any examples of failure cases. Without such analysis, it is unclear why P³ performs better—whether due to improved reasoning or merely longer, stylistically refined responses.
5. No enough ablation for hyper parameter. For example, α in expression (3) controlling the trade-off between persuasiveness and surface cues, yet no ablation or sensitivity study is provided to justify its selection or stability.
6. Lack of clear definition of persuasiveness: The paper does not explicitly define what builds persuasiveness (e.g., length, strength of style, number of supporting arguments), making it unclear what aspects the model actually learns or optimizes.

**Questions:**

1. Clarification on Figure 2
The paper would benefit from a clearer explanation of Figure 2. Specifically, please specify what the x- and y-axes represent in the scatter plot, and clarify the interpretation of the plotted metrics. It would also help to explicitly state whether higher values indicate better performance for each metric .

2. Bias Mitigation
The paper briefly mentions several sources of bias but does not provide sufficient methodological detail on how they are addressed.
(i) Data bias (Reddit): Please elaborate on how cultural and popularity biases are mitigated when training the MLPs, given that popularity signals (upvotes, downvotes) are not equivalent to genuine persuasiveness.
(ii) Accumulated bias: The training pipeline may compound multiple biases — Reddit community bias → flawed supervision signals → MLP learns spurious correlations → biased reward f_d → misguided PAPO optimization.  It would strengthen the work to clarify what specific mechanisms are applied to break or mitigate this propagation.
(iii) Evaluation bias: Since o1_score relies on GPT-based judgments, can you do some statistic analysis to show o1_score is diverse and would not give high score for some specific topic? What's could you please explain how to reduce the bias in the human evaluator during your experiments.

---

> ### Author Response · Authors · 2025-11-20
>
> ### Thank you for your careful review. We really appreciate your efforts in reviewing our paper.
>
> > The paper’s exposition is often unclear and difficult to understand.
>
> Thank you for your suggestion. We will include additional clarifications on technical details in the final version of the paper. If you have any questions regarding these details, please feel free to raise them during the discussion phase. We will address your concerns and questions one by one in the follow-up discussions.
>
> > The experiments compare P³ mainly with general instruction-tuned LLMs (e.g., LLaMA, Claude, Gemini), but omit specialized baselines that also target persuasion or argumentation, such as: SFT-based methods, DPO or similar methods are focusing on reward hacking and reward sparsity.
>
> It should be noted that the closed-source commercial models involved in this study have also been trained on CMV; for example, the GPT-4o1 technical report explicitly includes training and evaluation on the CMV task [1]. Some models, such as Claude, have been specifically optimized for persuasion-related tasks [2], making our comparisons fair. Naturally, as you suggested, including additional baselines trained on this task helps better demonstrate the practical performance of our method. Therefore, we added the PESA baseline [3], whose training objective also focuses on enhancing model persuasiveness. Experimental results show that our approach significantly outperforms this strong baseline on persuasion metrics. Due to space limitations, we will include these results in the supplementary response.
>
> [1] Openai o1 system card
>
> [2] Measuring the Persuasiveness of Language Models: https://www.anthropic.com/news/measuring-model-persuasiveness
>
> [3] Prove Your Point!: Bringing Proof-Enhancement Principles to Argumentative Essay Generation
>
> > The paper reports only the O₁-score (GPT-4 evaluation) and limited human judgments. Even though some previous metrics have limitation, it can reflect something by using Bert score, Rouge/BLEU. Such as semantic alignment and style or phrasing pattern.
>
> The reason we do not use ROUGE or BLEU is that they only capture semantic overlap at the word level, which fails to assess the persuasiveness of the text. In fact, as shown in Figure 3 of the experimental analysis section, the Pearson correlation coefficients between ROUGE and BLEU scores and human judgments of persuasiveness are both below 0.1. This is because persuasiveness arises from more nuanced logical relationships, which can be drastically affected by minor changes in wording. Similarly, semantic similarity metrics based on pre-trained models, such as BERTScore, are also unsuitable for the same reasons as BLEU and ROUGE. In contrast, the automatic evaluation metric we employ, O1-Score, exhibits a correlation of approximately 0.6 with true persuasiveness, which we consider sufficiently accurate to reflect the persuasive quality of the text.
>
> > The paper lacks both statistical and qualitative analyses of generated outputs.
>
> We will include additional statistical analyses of the outputs in the final version. To address your concerns regarding the source of P3’s effectiveness, we conducted supplementary experiments analyzing the length and diversity of P3-generated texts. The results show that P3’s outputs are slightly longer than those of other baseline models, possibly due to generating more evidence and logical reasoning steps. At the same time, P3’s diversity metrics do not differ significantly from other baselines, indicating that its persuasiveness does not stem from a particular writing style.
>
> > No enough ablation for hyper parameter. For example, α in expression (3) controlling the trade-off between persuasiveness and surface cues, yet no ablation or sensitivity study is provided to justify its selection or stability.
>
> Thank you for your suggestion. We conducted additional ablation experiments on the parameter α, which show that the O1-score exhibits a unimodal pattern, achieving optimal performance around α = 0.6. This indicates that in actual Reddit discussions, only a minority of the audience evaluates based on textual persuasiveness, while the majority rely more on superficial features such as writing style.

---

> > ### Author Response · Authors · 2025-11-20
> > **Online debate supplementary experiment**
> >
> > In this response, we will provide the results on out-domain online debate dataset DDO corpus (mentioned in [1] and [2]). Due to the time limitations, we only select 100 test samples and three strong baselines. We will add full results in the final version of the paper.
> >
> > [1] Exploring the Role of Prior Beliefs for Argument Persuasion
> >
> > [2] A Corpus for Modeling User and Language Effects in Argumentation on Online Debating
> >
> > | Model | o1-score | human-eval |
> > |------|------------|---------------|
> > | Qwen2-13B | 75.62 | 4.30 |
> > | Qwen2-72B | 77.35 | 4.42 |
> > | Gemini1.5 Flash | 80.21 | 4.66 |
> > | Ours | **83.32** | **4.73** |

---

> > ### Comment · Reviewer_ChvZ · 2025-11-21
> >
> > I appreciate the detailed responses, which have clarified most of my concerns. However, I would like to clarify my previous request regarding the results analysis, as I apologize that I may not have been explicit enough.
> >
> > While I acknowledge the limitations of metrics like BLEU, I requested them (or similar automatic metrics) because the current O1 score is too general or high-level to fully capture the nuances of 'persuasiveness.'(You can use any other metrics if they can well explain detailed persuasiveness. This may show at what aspect your method perform best.) To robustly demonstrate effectiveness, granular automatic metrics are needed to explain the high-level scores.
> >
> > Therefore, to strengthen the paper, it is essential to include multi-dimensional metrics (e.g., repetitiveness, evidence usage, reasoning, and language style, similar to [1]) and statistical analysis regarding length and evidence counts. A single case study in supplementary is insufficient to generalize the model's performance.
> >
> > Given the time constraints, could you please provide a comparison of your method against PESA and Gemini 1.5 based on the detailed metrics used in [1] or use some other metrics to explain your o1 score? (You can test on a sub-dataset) If you can successfully demonstrate consistent improvements on these granular indicators and add them into your final version, I am willing to raise my score.
> >
> > [1]. Prove Your Point!: Bringing Proof-Enhancement Principles to Argumentative Essay Generation

---

> > > ### Author Response · Authors · 2025-11-21
> > >
> > > Thank you for your fast and clear response. Your insights are highly constructive. Although the current O1 score can evaluate the overall persuasiveness of the text, it indeed lacks finer-grained assessment. Therefore, we conducted supplementary experiments by sampling 1,000 test instances and, using the same evaluation prompt as PESA, assessed the following five metrics employed in this paper: Relevance, Validity of Reasoning, Credibility of Evidence, Language and Rhetoric, and Overall Persuasiveness. The experimental results are as follows.
> > >
> > > | Model | Relevance | Validity of the Reasoning | Credibility of Evidence | Language and Rhetoric | Overall Persuasiveness |
> > > |------|------------|------------|------------|------------|------------|
> > > | Gemini 1.5 | **91.75** | 85.32 | 50.51 | **80.33** | 78.10 |
> > > | PESA | 90.54 | 83.97 | 49.13 | 70.69 | 75.20 |
> > > | Ours | 90.72 | **88.56** | **64.49** | 63.91 | **80.17** |
> > >
> > > The experiments demonstrate that the P3 method achieves improvements over two strong baselines in Validity of Reasoning, Credibility of Evidence, and Overall Persuasiveness. This indicates that P3 effectively captures, trains on, and optimizes key elements of argumentative persuasiveness in weakly supervised text, thereby enhancing the logical coherence and evidential strength of the generated outputs. On the Relevance metric, P3 performs slightly below the baselines, likely due to uneven topic distribution in the Reddit data, which may have somewhat hindered the model’s topic understanding. P3 shows weaker performance on the Language and Rhetoric metric, substantially lower than both baselines; this is because P3 treats linguistic style and related features as surface cues (S_s) unrelated to persuasiveness during modeling, thereby down-weighting their influence on the generation model’s language and rhetorical aspects during training. Overall, the results align with expectations for P3’s performance on fine-grained persuasiveness evaluation. We will include the full experimental results on all test data and comprehensive human evaluation experiments in the final version.

---

> ### Author Response · Authors · 2025-11-20
>
> > Lack of clear definition of persuasiveness
>
> Several linguistic and theoretical studies on argumentation indicate that the validity of logical reasoning is the primary source of persuasiveness [1][2]. The audience’s beliefs do influence persuasion, and our method accounts for this aspect as well. On one hand, during the modeling process in Persuasiveness Reward Estimation and Modeling, we probabilistically model audience behavior that may be misled by superficial cues when expressing approval, estimating the model parameters using the EM algorithm. On the other hand, the weak supervision signal “score” we employ is derived from audiences with diverse beliefs, stances, and traits, thereby mitigating subjective differences in persuasiveness evaluation at the individual level to some extent.
>
> [1] Why we must argue: A critique of the essence, purpose, and craftsmanship
> of argumentation.
>
> [2] Toulmin's model of argumentation.
>
> > Clarification on Figure 2 The paper would benefit from a clearer explanation of Figure 2
>
> In the scatter plot of Figure 2, the x-axis represents the values of the respective metrics in the title (for example, BLEU scores for the BLEU plot), while the y-axis represents the O1-score (indicating textual persuasiveness). This figure primarily illustrates the limitations of BLEU and ROUGE in evaluating persuasiveness, as discussed in the rebuttal above.
>
> > Bias Mitigation The paper briefly mentions several sources of bias but does not provide sufficient methodological detail on how they are addressed.
>
> Our method addresses or mitigates the three types of biases you mentioned.
>
> For data bias, our DPO algorithm simultaneously optimizes the generated text’s “score” (defined as the difference between upvotes and downvotes) and its “debate score” (estimated during the Persuasiveness Reward Estimation and Modeling step). The goal is to enhance text persuasiveness through this joint optimization, and experimental results demonstrate its effectiveness. As shown in Figure 3, the extracted debate score \(S_d\) correlates strongly with human persuasiveness ratings (approximately 0.5), thereby addressing the bias present in the original score.
>
> Regarding evaluation bias, the consistency with human annotations validates the effectiveness of the O1-score. We will include an analysis of O1-score across different topics in the final version to further substantiate this point.
>
> For cumulative bias, the PAPO process explicitly accounts for potential deviations arising from persuasiveness and reward estimation. First, we use the relative magnitude of persuasiveness scores as the optimization objective, which mitigates the high variance issue in persuasiveness estimation. Second, during PAPO, we incorporate a smoothing coefficient based on the debate score differences, so that samples with larger debate score gaps contribute more to the final outcome. This further prevents the DPO algorithm from learning erroneous preferences due to variance in persuasiveness estimation. Although PAPO may introduce new errors, these mitigation strategies prevent exponential accumulation of such errors.

---

> ### Author Response · Authors · 2025-11-20
> **Supplementary Hyperparameter Experiment**
>
> In this response, we present the effect of the hyperparameter alpha on the O1-score.
>
> | Alpha | o1-score |
> |------|------------|
> | 0.4 | 70.23 |
> | 0.5 | 75.62 |
> | 0.6 | **78.29** |
> | 0.7 | 77.20 |
> | 0.8 | 75.01 |

---

> ### Comment · Reviewer_ChvZ · 2025-11-26
>
> I appreciate the authors’ effort in providing extensive additional experiments and clarifications during the rebuttal, which helped address many of my initial concerns. Based on this improved understanding, I am increasing my score from 2 to 4.
>
> However, consistent with reviewer mPUe’s comments and my own earlier confusion during the initial reading, I believe the presentation quality still requires improvement. In particular, some parts of the paper could be made clearer and more accessible to readers on the first pass.
>
> Therefore, I strongly recommend that the authors refine the paper’s presentation and incorporate the analyses and insights introduced during the rebuttal phase. As noted in the official guidelines (https://iclr.cc/FAQ/RevisePaper), submitting a revised version following these improvements would benefit the final submission and its readability.

---

### Note · Authors · 2025-12-03

I have read and agree with the venue's withdrawal policy on behalf of myself and my co-authors.